# Catalytic atroposelective synthesis of axially chiral benzonitriles via chirality control during bond dissociation and CN group formation

Ya Lv[1], Guoyong Luo[2], Qian Liu[1], Zhichao Jin[1✉], Xinglong Zhang [3✉] & Yonggui Robin Chi [1,4✉]

The applications of axially chiral benzonitriles and their derivatives remain mostly unexplored due to their synthetic difficulties. Here we disclose an unusual strategy for atroposelective access to benzonitriles via formation of the nitrile unit on biaryl scaffolds pre-installed with stereogenic axes in racemic forms. Our method starts with racemic 2-arylbenzaldehydes and sulfonamides as the substrates and N-heterocyclic carbenes as the organocatalysts to afford axially chiral benzonitriles in good to excellent yields and enantioselectivities. DFT calculations suggest that the loss of *p*-toluenesulfinate group is both the rate-determining and stereo-determining step. The axial chirality is controlled during the bond dissociation and CN group formation. The reaction features a dynamic kinetic resolution process modulated by both covalent and non-covalent catalytic interactions. The axially chiral benzonitriles from our method can be easily converted to a large set of functional molecules that show promising catalytic activities for chemical syntheses and anti-bacterial activities for plant protections.

[1] State Key Laboratory Breeding Base of Green Pesticide and Agricultural Bioengineering, Key Laboratory of Green Pesticide and Agricultural Bioengineering, Ministry of Education, Guizhou University, Huaxi District, Guiyang 550025, China. [2] School of Pharmacy, Guizhou University of Traditional Chinese Medicine, Huaxi District, Guiyang 550025, China. [3] Institute of High Performance Computing, A*STAR (Agency for Science, Technology and Research), Singapore 138632, Singapore. [4] Division of Chemistry & Biological Chemistry, School of Physical & Mathematical Sciences, Nanyang Technological University, Singapore 637371, Singapore. ✉email: zcjin@gzu.edu.cn; Zhang_Xinglong@ihpc.a-star.edu.sg; robinchi@ntu.edu.sg

Benzonitrile is a unique motif in functional molecules and organic synthesis. To date, these benzonitrile groups have mostly been prepared as part of a larger molecule that contains central chiralities[1]. Indeed, such carbonitrile motifs have appeared in many centrally chiral molecules with proven applications as medicines, such as Finrozole[2], Osilodrostat[3,4], and Cromakalim[5,6] (Fig. 1a). For example, the benzonitrile group of the breast cancer-treating drug Finrozole acts as a hydrogen bond acceptor to inhibit the aromatase enzyme of P450 arom[1]. The benzonitrile unit of the FDA-approved drug Osilodrostat plays important roles in inhibiting human aldosterone synthases CYP11B2 via aromatic interactions with the Trp116 side chain[3]. Cromakalim containing a benzonitrile fragment with multiple chiral centers is the first-generation drug that was used to cure hypertension[5]. In sharp contrast, the preparation of axially chiral molecules bearing benzonitrile units with high optical purities remains challenging[7,8]. Possible methods to address this challenge may involve the formation of the stereogenic $C(sp^2)$-$C(sp^2)$ axis (Fig. 1b)[9–11]. Unfortunately, effective control of the atropoenantioselectivity in the formation of the stereogenic $C(sp^2)$-$C(sp^2)$ axis of the axially chiral benzonitrile products has not been realized. Consequently, indirect methods with the use of the

stoichiometric amount of chiral reagents are required in the current synthesis of axially chiral benzonitriles from commercial starting materials[7,8]. This synthetic challenge has made it very difficult to explore axially chiral benzonitriles and their derivatives for applications. It becomes clear to us that a strategy with other modes of chirality control is needed in order to achieve rapid preparation of axially chiral benzonitriles and their derivatives. As a result, our attentions move beyond the $C(sp^2)$-$C(sp^2)$ axis to the C≡N carbonitrile units.

In this work, we set up the axial chirality through a catalytic C≡N triple bond formation process (Fig. 1b)[12–16]. Our approach involves a single-pot operation to prepare 2-arylbenzonitriles with up to quantitative yields and excellent optical purities (Fig. 1c). We use a racemic mixture of 2-arylbenzaldehydes (1) and readily available sulfonamides ($RSO_2NH_2$, 2) as substrates and an aminoindanol-derived N-heterocyclic carbene (NHC)[17–24] as an organic catalyst. A hydroxyl group is installed to the 2-arylbenzaldehyde substrate to promote chirality control and reaction efficiencies via intramolecular non-covalent interactions in a key catalyst-bound intermediate, as elucidated via DFT calculations. The observed high optical purities of the products are achieved through a sophisticated catalytic dynamic kinetic resolution (DKR) process. The chirality of the final product is believed to be established during the N-S bond-breaking step before the formation of the C≡N triple bond. Our axially chiral biaryl products contain both a carbonitrile and a hydroxyl group that are both useful in many settings. For example, both CN[25–27] and OH[28–32] groups in our products can directly serve as ligands for transition metals in asymmetric catalysis. These groups can also be easily transformed into other moieties such as amines and carboxylic acids with a broad presence in functional molecules such as medicines. We demonstrate that multiple chiral catalysts/ligands can be prepared from our products using straightforward operations. In a broader perspective, we expect our strategy to open alternative avenues for chiral inductions and asymmetric catalysis by designing proper bond dissociation processes.

## Results

**Reaction development.** The racemic mixture of the 2-arylbenzaldehyde **1a** bearing a stereogenic axis and the toluenesulfonamide **2a** were selected as the model substrates to search for a suitable reaction condition for the atropoenantioselective DKR process (Table 1). To our great delight, the enantiomerically enriched axially chiral benzonitrile product **3a** was smoothly formed with promising er values using various indanol-derived NHC catalysts (Table 1, entries 1 to 3)[33–36]. Thereafter, modified NHC precatalysts bearing a $NO_2$ group on their indanol moieties were evaluated in order to further improve the product optical purities (entries 4 to 7). NHC pre-catalysts **D**[37] and **E**[38] bearing electron-rich N-substituents gave the benzonitrile product **3a** in only moderate yields (entries 4 and 5), while NHC pre-catalyst **F**[39] with an electron-deficient N-$C_6F_5$ group gave the desired product in an excellent yield with an increased optical purity (entry 6). The product yield could be further improved with retention of the reaction enantioselectivity when switching the conjugate anion of the NHC pre-catalyst **F** from $BF_4^-$ to $Cl^-$ (entry 7, NHC precatalyst **G**).

The use of different bases has significant impacts on both the product yields and er values (entries 8 to 13). The inorganic bases other than $Cs_2CO_3$ (e.g., entries 8 and 9) and various tertiary amines (e.g., entries 10 and 11) gave the desired product **3a** in excellent enantioselectivities, but only with moderate yields. Interestingly, secondary amines such as $NH(C_2H_5)_2$ were found to be extraordinarily beneficial for this DKR transformation,

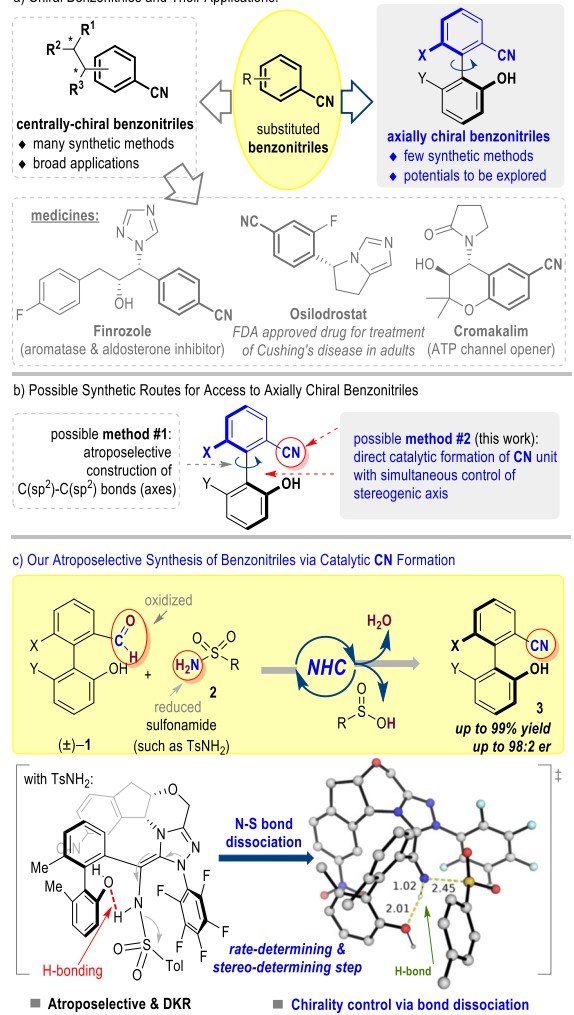

a) Chiral Benzonitriles and Their Applications:

**centrally-chiral benzonitriles**
♦ many synthetic methods
♦ broad applications

substituted **benzonitriles**

**axially chiral benzonitriles**
♦ few synthetic methods
♦ potentials to be explored

medicines:

**Finrozole**
(aromatase & aldosterone inhibitor)

**Osilodrostat**
*FDA approved drug for treatment of Cushing's disease in adults*

**Cromakalim**
(ATP channel opener)

b) Possible Synthetic Routes for Access to Axially Chiral Benzonitriles

possible **method #1**: atroposelective construction of $C(sp^2)$-$C(sp^2)$ bonds (axes)

possible **method #2** (this work): direct catalytic formation of **CN** unit with simultaneous control of stereogenic axis

c) Our Atroposelective Synthesis of Benzonitriles via Catalytic **CN** Formation

oxidized
reduced sulfonamide (such as TsNH₂)

(±)–**1**

**NHC**

up to 99% yield
up to 98:2 er

**3**

with TsNH₂:

N-S bond dissociation

H-bonding

*rate-determining & stereo-determining step*

H-bond

■ Atroposelective & DKR
■ Metal-free CN formation
■ Chirality control via bond dissociation
■ Versatile benzonitrile products

**Fig. 1 Applications of benzonitriles and their syntheses. a** Chiral benzonitriles and their applications. **b** Possible synthetic routes for access to axially chiral benzonitriles. **c** Our atroposelective synthesis of benzonitriles via catalytic CN formation.

**Table 1 Optimization of reaction conditions[a].**

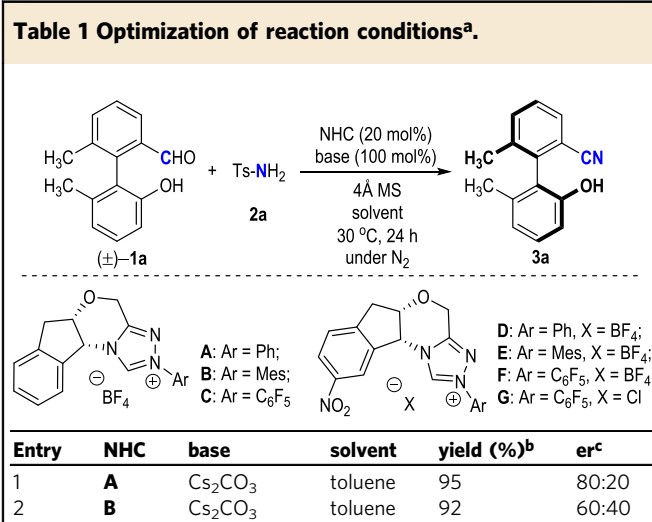

| Entry | NHC | base | solvent | yield (%)[b] | er[c] |
|---|---|---|---|---|---|
| 1 | A | Cs$_2$CO$_3$ | toluene | 95 | 80:20 |
| 2 | B | Cs$_2$CO$_3$ | toluene | 92 | 60:40 |
| 3 | C | Cs$_2$CO$_3$ | toluene | 91 | 72:28 |
| 4 | D | Cs$_2$CO$_3$ | toluene | 65 | 89:11 |
| 5 | E | Cs$_2$CO$_3$ | toluene | 65 | 60:40 |
| 6 | F | Cs$_2$CO$_3$ | toluene | 89 | 90:10 |
| 7 | G | Cs$_2$CO$_3$ | toluene | 97 | 90:10 |
| 8 | G | NaOH | toluene | 41 | 96:4 |
| 9 | G | K$_2$CO$_3$ | toluene | 46 | 96:4 |
| 10 | G | DABCO | toluene | 49 | 97:3 |
| 11 | G | N(C$_2$H$_5$)$_3$ | toluene | 57 | 97:3 |
| 12 | G | NH(C$_2$H$_5$)$_2$ | toluene | 96 | 98:2 |
| 13 | G | DBU | toluene | 82 | 79:21 |
| 14 | G | NH(C$_2$H$_5$)$_2$ | THF | 62 | 91:9 |
| 15 | G | NH(C$_2$H$_5$)$_2$ | CH$_2$Cl$_2$ | 93 | 94:6 |

[a]Unless otherwise specified, the reactions were carried using **1a** (0.10 mmol), TsNH$_2$ (0.11 mmol), NHC (0.02 mmol), base (0.10 mmol), 4 Å MS (50 mg), and solvent (1.0 mL) at 30 °C under N$_2$ for 24 h.
[b]Isolated yield of **3a**.
[c]The er values of **3a** were determined via HPLC on the chiral stationary phase.

affording the target chiral benzonitrile **3a** in an excellent yield and optical purity (e.g., entry 12). It is worth noting that the product er value dropped substantially when using DBU as the base (entry 13), possibly due to the interruption of the existing non-covalent interactions between the NHC catalyst and the imine intermediates. The reaction could also be carried out in a variety of organic solvents, though the yields and optical purities of the products were a bit lower (e.g., entries 14 and 15).

**Reaction scope**. Having obtained an optimal reaction condition for this dehydrative DKR transformation (Table 1, entry 12), we then examined the substrate scope for the atroposelective synthesis of axially chiral benzonitriles (Table 2). Electron-donating groups are well tolerated on each position of the **A** rings of the benzaldehydes **1**, with the axially chiral biphenyl carbonitriles **3** afforded in good to excellent yields and enantioselectivities (**3b** to **3d**, **3g** to **3i**). However, incorporating an electron-withdrawing Cl atom on the 6-methyl-2-hydroxylphenyl ring **A** leads to drops of the product er value (**3f**, **3j**). The 6-methyl group on the phenol ring **A** that accounts for rotational barriers can be switched to a methoxyl or Cl group, with the corresponding benzonitrile products afforded in good yields with excellent optical purities (**3e**, **3k**, and **3l**).

Both electron-donating and electron-withdrawing substituents can be installed on the 2-formylphenyl **B** rings of the benzaldehydes **1** to give various substituted axially chiral benzonitriles in good to excellent yields and enantioselectivities (**3m** to **3t**). 2-Naphthylcarbaldehydes also worked well in this

NHC-catalyzed atropoenantioselective DKR process, with the target axially chiral naphthyl-2-carbonitriles afforded in excellent yields with good to excellent optical purities (**3u** to **3y**).

It is worth noting that slight alterations of the reaction conditions are sometimes needed in the preparations of diversely substituted axially chiral benzonitrile derivatives. The optical purities of the benzonitriles **3** can be effectively improved with 1-naphthylphenol as additives in these cases. Moreover, the atroposelective protocol can also be carried out at big scales without obvious erosions on the reaction yields and enantioselectivities (e.g., Tables 2, 3a at 1 mmol scale).

The OH groups that existed on the **A** rings of the reaction substrates **1** were found to be crucial for the NHC-catalyzed nitrile formation process. The reaction using the benzaldehyde substrate where the 2-OH group on the **A** ring is replaced by 2-methoxy group gave the target product only in trace amount (e.g., **3z**). The 2-OH group of the **A** ring was theoretically found to stabilize the transition state structure and lower the barrier of the rate-determining transition state in the present transformation. Moreover, the 2-OH group was essential for the formation of the bridged hemiacetal or hemiaminal intermediates to ensure the racemization process for the DKR transformation (*vide infra*).

A variety of substituted sulfonamides **2** can be used as the N sources instead of the tosyl amide (TsNH$_2$) for the asymmetric benzonitrile synthesis. The 4-methyl group on TsNH$_2$ can be replaced with either an electron-donating or an electron-withdrawing group without much erosion of the product's optical purities (**2b** to **2d**). However, substitutions on the *o*-positions of the phenyl groups of the sulfonamides resulted in significant drops of the product er values, although the yields were not affected (**2e** and **2f**). The aromatic groups on the sulfonamide substrates can also be replaced with alkyl groups, but they only gave product **3a** with decreased yields and optical purities under the current catalytic conditions (**2g** and **2h**). Switching the sulfonamide substrates **2** into alkyl-/arylamines or benzamides leads to no formation of the target benzonitrile products.

**Mechanistic study**. To understand the mechanism of the present transformation, density functional theory (DFT) calculations were performed (DFT calculations were performed with ORCA 5.0.1 and Gaussian 16 Rev B.01 software at SMD (toluene)-DLPNO-CCSD(T)/cc-pV(DT)Z CBS Extrapolation//M06-2X/def2-SVP level of theory. See Supplementary Information for further details.). We carried out a model calculation in which a model NHC and a model imine are used to initially explore the potential energy surface of this reaction (Fig. 2, see Supplementary Information for further details). Note that, for the **model_NHC** we used, the reaction center is similar to the chiral NHC catalyst used in the real catalytic reaction. For the imine simplification, we note that the methanesulfinate group (**model_imine** in Fig. 2) has similar reactivity as the p-toluenesulfinate group. We, therefore, use this model reaction to determine the key steps for the overall transformation, from which we applied the full model to the key step to determine the stereoselectivity. We found that the reaction preceded with firstly the addition of NHC catalyst to the imine C=N bond, giving a highly exergonic adduct **model_INT2**, at −9.3 kcal mol$^{-1}$. This is followed by the loss of methanesulfinate anion, via transition state **model_TS2**, at 23.9 kcal mol$^{-1}$. The final deprotonation of imine intermediate via **model_TS3** regenerates the NHC catalyst and yields the carbonitrile product.

We herein focus on the steps of NHC addition and the loss of methanesulfinate since these steps are likely stereo-determining in the overall transformation of the full system as the regeneration of NHC catalyst via **model_TS3** through deprotonation is likely facile and simply carries the stereochemical information from

**Table 2 Substrate scope of the atroposelective benzonitrile synthesis[a].**

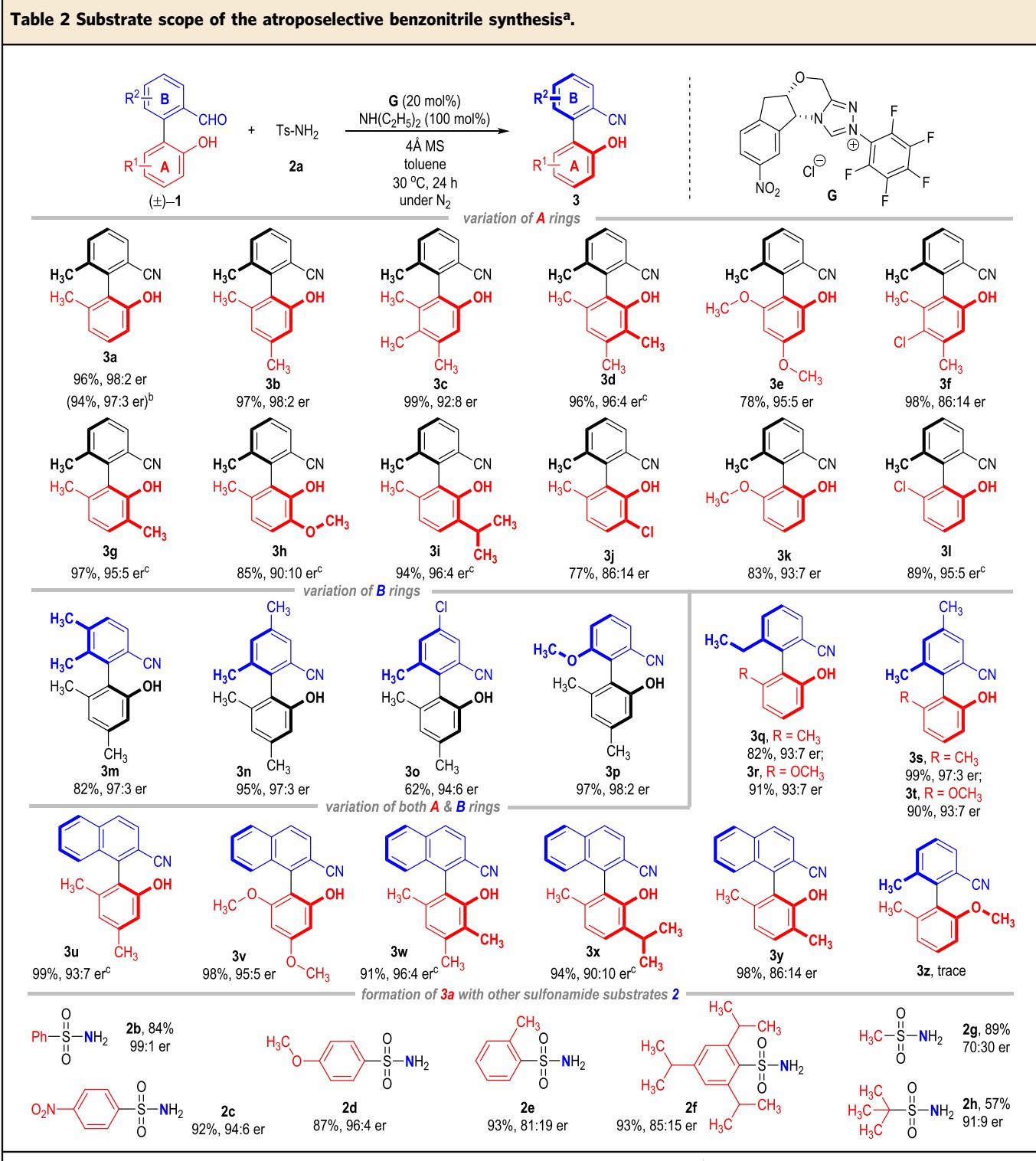

previous steps forward. From the Gibbs energy profile in Fig. 2, we can see that the NHC adduct, **model_INT2**, is the resting state of the catalytic cycle. The rate-limiting step is the loss of methanesulfinate, **model_TS2**, with an energetic span of 33.2 kcal mol⁻¹ (from **model_INT2** to **model_TS2**). Moreover, the addition of NHC, **model_TS1**, is reversible, as the subsequent loss of methanesulfinate has a barrier of 33.2 kcal mol⁻¹, which is

higher than the barrier for the reversible process of adduct dissociation (going from **model_INT2** to **model_INT1**) with a barrier of 24.4 kcal mol⁻¹. With these results, we focus on the step of loss of anion in the full system as both the rate-limiting and stereo-determining step.

For the full system, we found that the reaction pathway leading to the major product via the transition state, **major_TS2**, has an

energetic span of 23.7 kcal mol⁻¹, whereas the energetic span for the minor product formation via **minor_TS2** is 27.2 kcal mol⁻¹ (Fig. 3a) and Supplementary Fig. 6). This barrier difference of 3.5 kcal mol⁻¹ translates to an enantiomeric excess of 99%, at an experimental temperature of 30 °C. This is in good agreement with experimental observations. In addition, the energetic span of 23.7 kcal mol⁻¹ for the major atropoenantiomer formation is consistent with excellent reactivity at the experimental temperature of 30 °C. With similar HOMO structures (see Supplementary Fig. 5), **major_TS2** is likely more favored due to the hydrogen

bonding formed between the 2-OH group on the substrate and the imine group of the aza-Breslow intermediate (Fig. 3a), thus hinting at the importance of the 2-OH group in atropoenantios-electivity control.

The stereochemistries of the afforded benzonitrile products **3** are extremely stable under thermodynamic conditions. For example, the er value of the axially chiral product **3a** does not change even after stirring for 12 h at 180 °C in mesitylene. In addition, the stereochemical stabilities of the benzonitrile product **3a**, carbaldehyde starting material **1a**, and the toluenesulfonimide intermediate **4a** were all evaluated through DFT calculations (Fig. 3b). The barriers for their isomerization are all well over 50 kcal mol⁻¹, indicating that these atropisomers do not racemize easily at the reaction temperature of 30 °C. However, it is possible for atropisomers of the carbaldehyde starting materials and the dehydrated toluenesulfonimide intermediates to easily intercon-vert into each other via the formations of bridged biaryl lactol intermediates **1a′** and **4a′** (Fig. 3c)[40–46]. As a technical note, the formation of the biaryl lactol intermediate **1a′** can be evidenced via ¹H NMR analysis on the basic solution of the 2-arylbenzaldehyde **1a** (see Supplementary Fig. 8 in the Supple-mentary Information for details).

Based on both the experimental and computational studies on the NHC-catalyzed dehydrative DKR reactions, a plausible reaction mechanism is postulated as depicted in Fig. 3c. After the condensation of racemic 2-arylbenzaldehyde (±)−**1a** with TsNH₂ to give a racemic mixture of imines (±)−**4a**, the reaction proceeds with firstly the reversible addition of NHC catalyst to the imine C=N bond, giving a mixture of highly exergonic aza-Breslow intermediates **I** and **III**. This is followed by the loss of the *p*-toluenesulfonyl anion from the aza-Breslow intermediate,

**Table 3 Inhibitive activities of the axial chiral compounds against *Xoo*[a].**

| Compound | *Xoo* inhibition rate (%) | |
|---|---|---|
| | 100 µg/mL | 50 µg/mL |
| (+)−**3a** (*S*) | 88.80 ± 3.80 | 45.12 ± 1.12 |
| (−)−**3a** (*R*) | 59.10 ± 3.05 | 38.63 ± 2.40 |
| (±)−**3a** (*rac*) | 77.14 ± 2.38 | 46.30 ± 3.00 |
| (+)−**3l** (*S*) | 63.10 ± 4.60 | 36.07 ± 1.08 |
| (−)−**3l** (*R*) | 95.71 ± 1.15 | 53.39 ± 0.46 |
| (±)−**3l** (*rac*) | 97.38 ± 0.15 | 46.07 ± 4.20 |
| (+)−**3q** (*S*) | 73.75 ± 1.77 | 40.48 ± 7.88 |
| (−)−**3q** (*R*) | 87.92 ± 5.78 | 47.14 ± 8.13 |
| (±)−**3q** (*rac*) | 67.08 ± 4.27 | 56.07 ± 1.71 |
| BT | 86.85 ± 0.96 | 47.26 ± 6.84 |
| TC | 60.95 ± 2.57 | 34.88 ± 4.26 |

*BT* bismerthiazol, *TC* thiodiazole copper.
[a]All data were average data of three replicates.

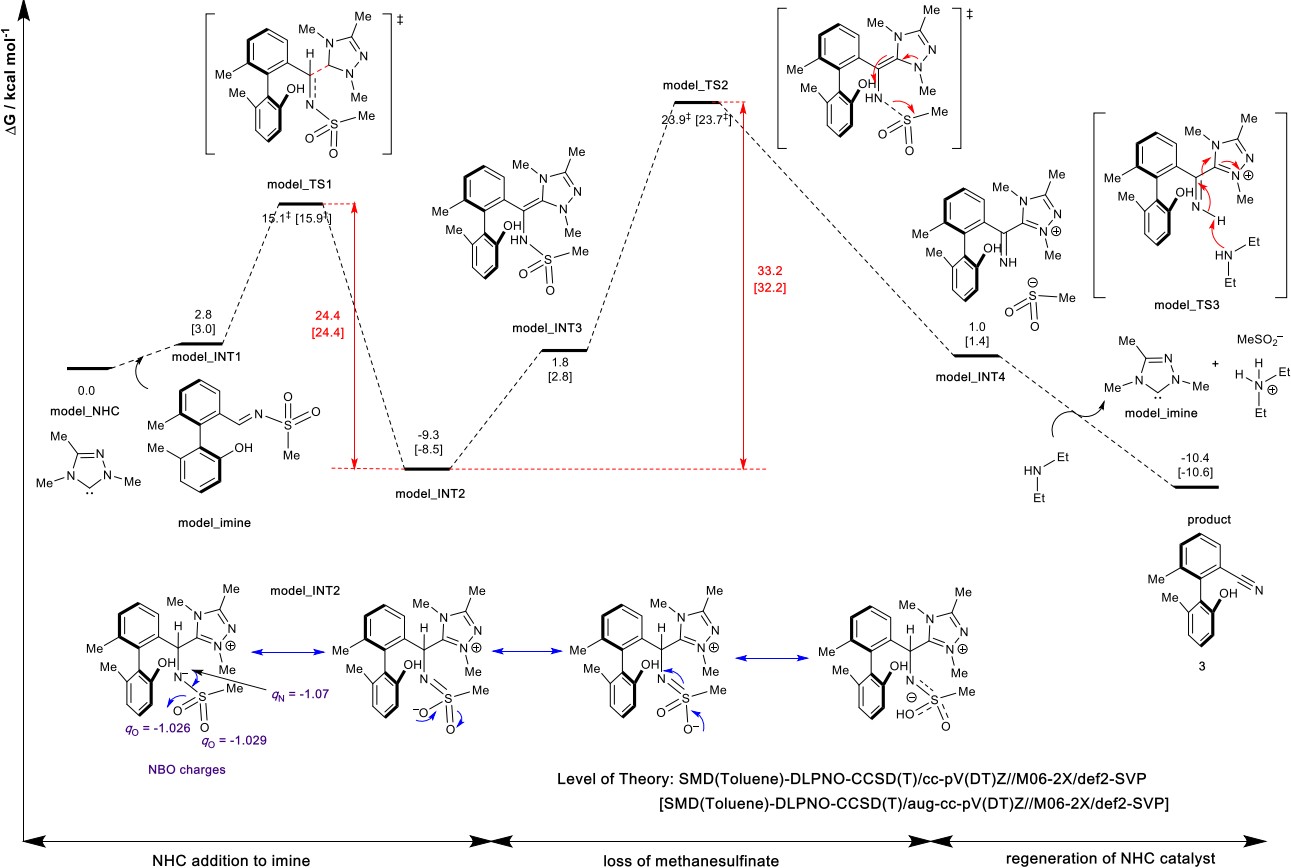

**Fig. 2 Gibbs energy profile for the model reaction system.** The rate-limiting step is the loss of methanesulfinate, **model_TS2**. The addition of NHC, **model_TS1**, is reversible. The step of loss of anion in the full system is both the rate-limiting and stereo-determining step.

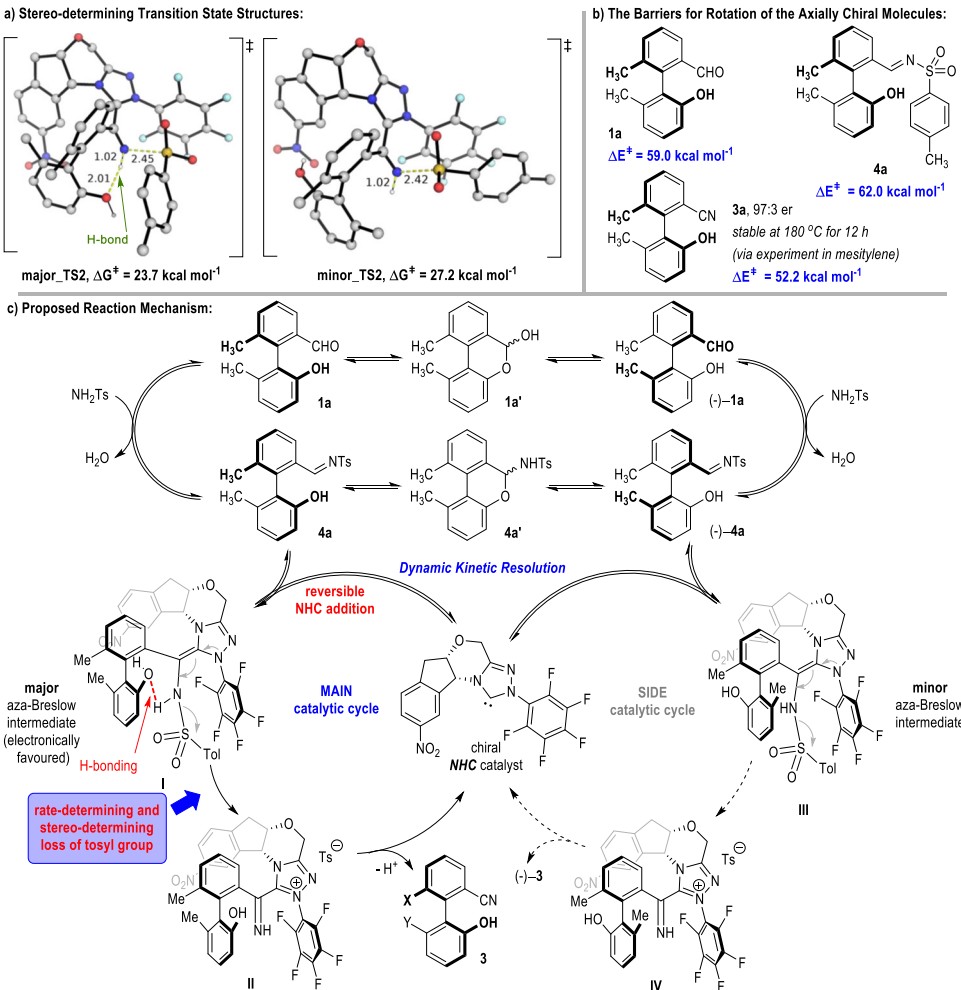

**Fig. 3 Proposed reaction mechanism for the atroposelective synthesis of benzonitriles. a** Stereo-determining transition state structures. **b** The barriers for rotation of the axially chiral molecules. **c** Proposed reaction mechanism.

which, having the highest reaction barrier for the overall transformation, is predicted to be rate-determining. This step is also stereo-determining as the addition of NHC catalyst to the condensed imine **4** is reversible and the subsequent deprotonation of intermediate **II/IV** simply carries the stereochemical information forward. Finally, deprotonation of the imine intermediate **II** regenerates the NHC catalyst and yields the carbonitrile product **3**.

**Synthetic applications of the afforded axially chiral benzonitrile derivatives.** Noteworthily, our presently revealed, first-reported enantioselective, transition metal-free preparations of axially chiral benzonitriles find diverse applications in asymmetric synthesis (Fig. 4). For example, the OH group of **3a** can be sulfonated with Tf$_2$O to give product **5** in an excellent yield without erosion on the optical purity (Fig. 4a). Compound **5** can be transformed to the multifunctional axially chiral phosphine **6** in good yield and enantioselectivity. The er value of the atropisomeric phosphine **6** can be further improved to >99:1 after recrystallization. Compound **6**, when reduced under a mild condition, gives the enantiomerically pure bifunctional aminophosphine **7** in good yield. The optical purity of **7** is believed to be excellent since it can produce the Boc protected amido-phosphine **8** in almost quantitative yield with >99:1 er value.

The CN group on **3a** can also be transformed into various functional groups. For instance, hydrolysis of the CN group leads

to the formation of the chiral amide **9** with little erosion of the product's optical purity. Thioamide **10** can be efficiently afforded from **3a** without erosion on the enantioselectivity. Reduction of the CN group gives the axially chiral primary amine **11** in good yield, which can be transformed to the multifunctional thiourea **12**, urea **13**, and square amide **14** in moderate to excellent yields with retention of the atropoenantioselectivities. The axially chiral sulfonamide **15**, another useful molecule with proven synthetic application in asymmetric synthesis, can be obtained in an excellent optical purity from the chiral primary amine **11** via simple protocols. Interestingly, axially chiral amidine **16** can also be efficiently obtained from the primary amine **11**. Hydrolysis of the amidine **16** leads to amide **17** in an excellent yield without loss of optical purity. Similarly, the optically enriched axially chiral amine **18** can be obtained from **3x**, and then gives the chiral amidine **19** in a good yield without much erosion on the enantioselectivity (since **20** was afforded from **19** in 97% yield, 96:4 er, Fig. 4b).

Many of the afforded axially chiral multifunctional molecules find promising applications in asymmetric synthetic chemistry (Fig. 4c). For example, the atropisomeric bifunctional biaryl **7** can be used as an effective ligand in the Pd-catalyzed enantioselective substitution reaction between the alkene **21** and the malonate **22**, with the chiral product **23**[47] afforded in a good yield and optical purity (Fig. 4c, eq. 1). The sulfonamide **15** can be used as an efficient ligand in the Lewis acid-catalyzed alkyl addition reaction of the aldehyde **24** and gives the chiral alcohol **25**[48] in an

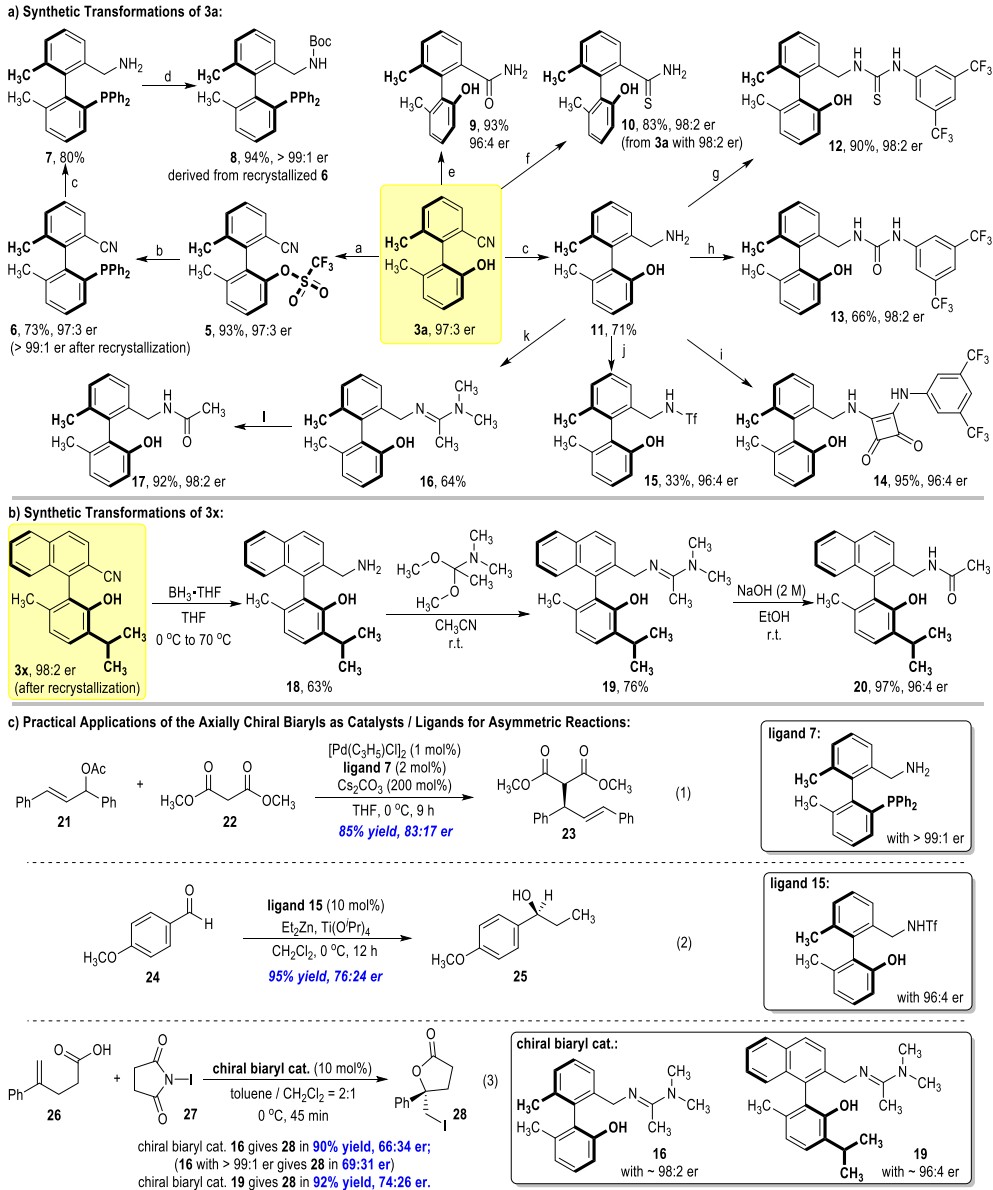

**Fig. 4 Synthetic transformations of the axially chiral benzonitriles products and their applications as chiral catalysts/ligands for asymmetric reactions.** [a]Tf$_2$O, pyridine, CH$_2$Cl$_2$, 0 °C to r.t. under N$_2$, 2 h; [b]HP(O)Ph$_2$, Pd(OAc)$_2$, dppb, DIEA and DMSO at 110 °C under N$_2$, 3 h; then HSiCl$_3$, DIEA and toluene at 100 °C under N$_2$, 2 h. [c]BH$_3$·THF, THF, 0 °C to 70 °C; [d](Boc)$_2$O, DMAP, CH$_2$Cl$_2$, r.t., 1 h; [e]Pd(OAc)$_2$, PPh$_3$, acetaldehyde oxime, EtOH/H$_2$O (v/v = 1/1), 70 °C, 3 h; [f]P$_2$S$_5$, EtOH, 70 °C, 24 h; [g]3,5-bis(trifluoromethyl)phenyl isothiocyanate, THF, r.t., 30 min; [h]3,5-bis(trifluoromethyl)phenyl isocyanate, THF, r.t., 30 min; [i]3-((3,5-bis(trifluoromethyl)phenyl)amino)-4-methoxycyclobut-3-ene-1,2-dione, MeOH, r.t., 30 min; [j]n-BuLi, Tf$_2$O, THF, −78 °C; then LiOH·H$_2$O, THF/H$_2$O (v/v = 10/3), r.t., 24 h; [k]1,1-dimethoxy-N, N-dimethylethan-1-amine, CH$_3$CN, r.t., 1 h; [l] NaOH (2 M), EtOH, r.t., 16 h. **a** Synthetic transformations of **3a**. **b** Synthetic transformations of **3x**. **c** Practical applications of the axially chiral biaryls as catalysts/ligands for asymmetric reactions.

excellent yield with a 76:24 er value (eq. 2). The chiral amidine **16** can be used as a promising organic catalyst for the enantioselective iodolactonization reaction of the alkene carboxylic acid **26** with NIS (**27**) to give the product **28**[8,49] in an excellent yield with moderate enantioselectivity (eq. 3). The er value of product **28** can be visibly improved when using the chiral amidine **16** with a higher optical purity. Adjusting the substitution patterns of the amidine **16** (e.g., the use of **19**) can also further increase the reaction enantioselectivity for the formation of **28**.

**Biological evaluations of the axially chiral benzonitrile products.** The benzonitrile moieties in our target products are believed to contribute to many bioactivities such as antimicrobial

activities in numerous studies. However, to the best of our knowledge, the bioactivities of axially chiral benzonitrile derivatives have not been studied. As part of our larger program to develop agricultural chemicals, here we evaluated our products in the inhibitive activities against *Xanthomonas oryzae* pv. *oryzae* (*Xoo*, Table 3)[50].

*Xoo* is a widespread bacterium that causes leaf blite in plants and results in large economic loss worldwide. Both of the optically enriched enantiomers of our axially chiral products were tested in vitro to inhibit *Xoo*, with the commercially available anti-bacterial thiodiazole copper (TC) and bismerthiazol (BT) being used as the comparison agents (positive controls). Racemic mixtures of the corresponding compounds were also evaluated. Distinctions in the inhibitive activities between the two

enantiomers of the same structures were observed. Some of our chiral products showed better results than the positive controls [Table 3, (+)−**3a**, (−)/(±)−**3l**, (−)−**3q**] at several concentrations and have shown promising applications in the development of agrichemical bactericides.

In summary, we have developed an atroposelective DKR reaction for the enantioselective preparation of axially chiral benzonitrile derivatives. Racemic mixtures of 2-arylbenzaldehydes bearing chiral axes and inexpensive sulfonamides are used as the reaction starting materials, with chiral NHC as the key organic catalyst. Atropisomeric benzonitriles bearing various substituents and substitution patterns are afforded in generally good to excellent yields and enantioselectivities. DFT calculations reveal that the loss of *p*-toluenesulfinate group as the leaving group is the key step for both rate- and stereo-determination, with the axial chirality well controlled during the bond dissociation and CN group formation event. The afforded chiral benzonitriles are rich in functionalities and can be transformed to a large set of atropospecific functional biaryls with broad applications as catalysts/ligands in asymmetric syntheses. Several of the axially chiral benzonitriles exhibit promising anti-bacterial activities against *Xoo* and can be used in the development of agrichemicals for plant protection. Further investigation into both the synthetic and the bioactive applications of the atropisomerically enriched benzonitriles is in progress in our laboratory and will be reported in due course.

## Methods

**General procedure for the enantioselective synthesis of 3**. To a 4.0 mL oven-dried vial equipped with a magnetic stir bar was added chiral NHC pre-catalyst **G** (0.02 mmol, 9.2 mg), 4 Å molecular sieves (50.0 mg), substrates **1** (0.10 mmol) and TsNH₂ **2a** (0.11 mmol). Then dried toluene (1.0 mL) and $NH(C_2H_5)_2$ (0.10 mmol, 10.3 μL) was added via syringe in a glove box under $N_2$ atmosphere. Then the reaction mixture was stirred for 24 h at 30 °C and then subjected to column chromatography on silica gel (10:1 petroleum ether/EtOAc) directly to give the desired pure products **3** in 62 to 99% isolated yields.

## Data availability

X-ray crystallographic data for compounds **3a**, **11** and NHC pre-catalyst **G** are available free of charge from the Cambridge Crystallographic Data Center under CCDC **2056374**, **2116213**, and **2093769**. Full experimental details for the preparation of all new compounds, and their spectroscopic and chromatographic data, can be found in the supplementary materials. Geometries of all DFT-optimized structures (in.xyz format) have been deposited with this Supplementary Information and uploaded to zenodo.org under open access (https://doi.org/10.5281/zenodo.5573970).

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

## Acknowledgements

We acknowledge financial support from the National Natural Science Foundation of China (21772029, 21801051, 21961006, and 22001173), The ten Talent Plan (Shicengci) of Guizhou Province ([2016]5649), the Science and Technology Department of Guizhou Province ([2019]1020, Qiankehejichu-ZK[2021]Key033), the Program of Introducing Talents of Discipline to Universities of China (111 Program, D20023) at Guizhou University, Frontiers Science Center for Asymmetric Synthesis and Medicinal Molecules, Department of Education, Guizhou Province [Qianjiaohe KY (2020)004], the Basic and Applied Research Foundation of Guangdong Province (2019A1515110906), the Guizhou Province First-Class Disciplines Project [(Yiliu Xueke Jianshe Xiangmu)-GNYL(2017) 008], Guizhou University of Traditional Chinese Medicine, and Guizhou University (China). Singapore National Research Foundation under its NRF Investigatorship (NRF-NRFI2016-06) and Competitive Research Program (NRF-CRP22-2019-0002); the Ministry of Education, Singapore, under its MOE AcRF Tier 1 Award (RG7/20, RG5/19), MOE AcRF Tier 2 (MOE2019-T2-2-117), MOE AcRF Tier 3 Award (MOE2018-T3-1-003); Nanyang Research Award Grant, Chair Professorship Grant, Nanyang Technological University. X.Z. acknowledges the support from IHPC, A*STAR, and thanks to the Deputy Chief Executive Research Office (DCERO), A*STAR for a Career Development Fund (CDF Project Number C210812008) for this work. X.Z. acknowledges the partial use of supercomputers in the A*STAR Computational Resource Center (ACRC) for computations performed in this work.

## Author contributions

Y.L. conducted most of the experiments. G.L. and Q.L. contributed to designs and some experiments. X.Z. conducted the DFT studies. Z.J. and Y.R.C. conceptualized and directed the project and drafted the manuscript with assistance from all coauthors. All authors contributed to part of the experiments and/or discussions. Y.L. and G.L. are estimated to contribute equally to this work.

## Competing interests

The authors declare no competing interests.
