## [Peer Review File · Nature Communications]

REVIEWER COMMENTS

Reviewer #1 (Remarks to the Author):

In this manuscript, Chi, Jin, Zhang and co-workers disclose an enantioselective approach for the synthesis of axially chiral benzonitrile derivatives through the efficient conversion of aldehyde to cyano group with NHC as the organocatalyst. Chiral nitrile molecules are of great significance due to their widespread applications in various fields. But the achievements in the construction were focused on central-chirality based structures. In this work, the first catalytic asymmetric synthetic method was developed through the rational introduction of the axial chirality, offering new and promising structures for the innovation of human medicines and agricultural chemicals.

Asymmetric Organocatalysis has proven to be a robust platform in constructing a variety of axially chiral structures, whereas the potential in the generation of carbonitrile groups still represents a challenging task in organic synthesis domain. Therefore, direct formation of nitriles from readily available and non-toxic starting materials is highly attractive, particularly in an enantioselective fashion. Undoubtedly, the method disclosed provides an elegant alternative strategy for the rapid access of the chiral nitriles. Moreover, this method features broad substrate generality, excellent efficiency and enantiocontrol. It is also pleasing to see that an in-depth study and discussion on reaction mechanism has been offered, followed by an impressive demonstration on the practical application of this methodology in both asymmetric catalysis and biological chemistry. Overall, this is a nice piece of work and should be of great interest in asymmetric axial chirality domain. It deserves publication on Nat. Comm. However, a few minor issues should be made before the publication.

1) This reviewer notices that the DFT study is carried out based on a model system that has not appeared in substrate expanding. The reason for the selection of the model system used in the DFT study should be clearly explained.

2) In the proposed mechanism, the authors explained that the starting materials racemize via the formation of bridged hemiacetal or hemiaminals. It is possible to give any experimental evidence on the intermediates? Also, can the imine intermediate be observed and isolated?

3) In page 3, the authors verified the necessity of OH group in the substrate (A ring rather than B ring, this should be corrected in the revised manuscript) through the reaction for the synthesis of 3z with a trace amount. They stated that "the 2-OH group was theoretically found to stabilise the transition state structure and". This author suggests that the 2-OH group was essential for the formation of the bridged hemiacetal or hemiaminal intermediate, then ensuring the racemization process. This point (also a feature of this design) should be included in the revised manuscript.

4) Only Tosyl type amines were evaluated. Did the authors have the results of any other amine?

5) What are the *er* values of axially chiral molecules (e.g., ligand 7, 15, 16, 19) used as catalysts in the asymmetric transformations depicted in Fig. 4? Can the optical purities of the corresponding products be further improved when using catalysts with higher enantiopurities? From 3a to 10, the enantiopurity was visibly increased. Could the authors give any explanation?

6) A quite similar work on atroposelective transformation of "Bringmann lactone" derivatives by Wang and co-workers should be cited (J. Am. Chem. Soc. 2016, 138, 6956–6959). Meanwhile, the first work on atropo-enantioselective ring opening of achiral biaryls by Bringmann and co-workers is suggested to be included (Angew. Chem. Int. Ed. 1992, 31, 761–762).

7) Other minor points to be corrected: Fig. 1, "central-chiral" to "centrally-chiral"; "versatile" to "versatile"; Fig. 2, the chemical bond was pointed to the middle of OH in some structures; please correct; Fig. 3b, for the investigation of the configurational stability of 3a, solvent "mesitylene" is suggested to be added.

Reviewer #2 (Remarks to the Author):

I believe that major revisions are necessary. There are a few inconsistencies between the proposed mechanism and the experimental findings that need to be addressed by the authors before the paper can be accepted for publication. Also, the authors should clarify some of the technical details used in their calculations. Please find enclosed a detailed report as a pdf.

The manuscript describes a novel reaction for the enantioselective synthesis of axially chiral benzonitrile derivatives. The mechanism of the transformation was investigated by means of DFT and DLPNO-CCSD(T) calculations. There are a few major inconsistencies between the proposed mechanism and the experimental findings that need to be addressed by the authors before the paper can be accepted for publication. Also, the authors should clarify some of the technical details used in their calculations. Please find below a list of major and minor remarks.

Major remarks:

- 1) The computed barrier of 33.1 kcal/mol for the loss of the methansulfinate anion (model_TS2) is too high to be reasonable under the reaction conditions. The authors could easily verify this by means of kinetic simulations, and should discuss in more detail the most likely error sources in their calculations (see points 3 and 4).
- 2) In Figure 2, the product is higher in energy than the intermediate model_INT2. If this is the case, what is the driving force for the reaction?
- 3) A possible error source in the calculations is the model system used by the authors in their preliminary study (Figure 2). To address this point, the authors should provide the reaction barrier for the loss of the methansulfinate anion computed using the full system. I could not find this number anywhere in the SI.
- 4) Another possible error source is basis set. The authors did not include any diffuse basis function in their study (I assume that this was done for efficiency reasons). As the key step of the reaction involves charged species (e.g., the methanesulfinate anion), this choice might lead to errors that are difficult to predict. The authors could verify this by computing (for the smaller model system) the reaction barrier with and without augmented functions.

Additional remarks:

- 1) In model_INT2, the negative charge is most likely delocalized over the oxygen atoms, while the positive charge is delocalized all over the ring. Thus, the Lewis shown in Figure 2 does not represent accurately the electronic structure of the intermediate.
- 2) The authors should clarify whether they have used the (T0) approximation in DLPNO calculations, which neglects the couplings between different triples by the off-diagonal Fock matrix elements, or the iterative algorithm (T1), which is also available for DLPNOCCSD(T).
- 3) The authors should clarify which PNO settings they have used for the DLPNO-CCSD(T) calculations (NormalPNO or TightPNO)
- 4) The authors should provide cc-pVDZ and cc-pVTZ values alongside with the extrapolated ones in the SI.

Reviewer #3 (Remarks to the Author):

Comments to NCOMMS-21-32504-T

Chi and co-workers describe in this manuscript a highly enantioselective protocol for atroposelective access to axial chiral biaryls containing benzonitrile moiety via catalytic formation of the nitrile (CN) functionality by using N-heterocyclic carbenes as the organocatalysts. As we know, axial chirality constitutes one unique chirality element which played essential role in life science as well as in material sciences. In the other hand, nitrile-containing compounds comprise a substantial proportion in the therapeutic drugs and the nitrile group represents a versatile functionality in organic synthesis. It is of great interest to install a nitrile group into atropisomerically chiral architectures based on catalytic asymmetric transformation. In this work, the authors focus on setting up the axial chirality through a catalytic C≡N triple bond formation reaction, using racemic mixture of 2-arylbenzaldehydes and readily available sulfonamides as substrates. The current protocol features fairly broad substrate generality, high efficiency and enantiocontrol under mild reaction conditions. Although the step involve the formations of bridged biaryl lactol intermediates for the catalytic dynamic kinetic resolution (DKR) process is relatively less prominent, the strategy for atroposelective formation of the C≡N triple bond remains very attractive and ingenious by comparison to existing methodologies. Moreover, the systematic investigation of mechanism, uniqueness of the target products as well as their application in asymmetric catalysis and antimicrobials should be highly prized as well. Therefore, this work should be considered as a significant improvement in related field and then I will give my support for publication in Nature Communication after addressing the following points:

- 1) Page 1, line 10-11, the sentence "Unfortunately, simultaneous control of the atropoenantioselectivity in such reactions has not been realised with current methods such as transition metal-catalysed asymmetric coupling reactions." is a bit confusing, please rephrase.
- 2) The stability of the represent axially chiral benzonitriles and its amine derivative (compound 11) is highly recommended to be investigated by reporting their half time.

- 3) This related work should be cited (Org. Lett. 2018, 20, 6284–6288).
- 4) SI, page 16 and page page 58, please control the amount of product and the corresponding yield.
- 5) For the supplementary: "additional X-ray crystallographic data for compound 11", inside Alert level C, the Friedel Pair Coverage was indicated to be too low (62%), which maybe not sufficient for the absolute configuration determination, it is recommended to conduct the X-Ray analysis by using copper target or other technologies.

Reviewer #1 has recommended acceptance of our manuscript after minor revisions.

- a) **Reviewer's comment:** "This reviewer notices that the DFT study is carried out based on a model system that has not appeared in substrate expanding. The reason for the selection of the model system used in the DFT study should be clearly explained."

Our Response: We have added the reason for the selection of the DFT model system in the revised manuscript on Page 3, Paragraph 5, as highlighted in yellow. In addition, this was elaborated in the Supplementary Information as "To initially explore the potential energy surface of this reaction and to increase computational efficiency, we carried out a model calculation in which a model NHC and a model imine is used (Supplementary Figure 3). Note that for the model NHC used, the reaction centre is similar as the chiral NHC catalyst used in the reaction. For the imine simplification, we note that the methanesulfonate group has similar reactivity as *p*-toluenesulfonate group. We use this model reaction to determine the key steps for the overall transformation, from which we applied the full model to the key step to determine the stereoselectivity."

- b) **Reviewer's comment:** "In the proposed mechanism, the authors explained that the starting materials racemize via the formation of bridged hemiacetal or hemiaminals. It is possible to give any experimental evidence on the intermediates? Also, can the imine intermediate be observed and isolated?"

Our Response: We have carried out control experiments trying to figure out the evidence on the generation of the hemiacetal intermediates (Figure R1).

The formation of the hemiacetal intermediate **1a'** can be observed via ¹H NMR analysis on the basic solution of the carbaldehyde **1a**. This information has been added in the revised Supplementary Information on Page 31 (Figure 8), and the revised manuscript on Page 5, Paragraph 2.

Figure R1. ¹H NMR analysis of the basic solution of the carbaldehyde **1a**.

The imine intermediates in this reaction turn out to be sensitive to moisture and are easily hydrolyzed. The instabilities of the imine intermediates make them difficult to be isolated and fully characterized. However, the formation of these imine intermediates can be observed via the HRMS analysis on the crude reaction mixture at the beginning of the reaction process (Figure R2). This information has been added in the revised Supplementary Information on Page 31 (Figure 9).

- c) Reviewer's comment: "In page 3, the authors verified the necessity of OH group in the substrate (A ring rather than B ring, this should be corrected in the revised manuscript) through the reaction for the synthesis of 3z with a trace amount. They stated that "the 2-OH group was theoretically found to stabilise the transition state structure and". This author suggests that the 2-OH group was essential for the formation of the bridged hemiacetal or hemiaminal intermediate, then ensuring the racemization process. This point (also a feature of this design) should be included in the revised manuscript."

Our Response: We have changed the "B ring" into "A ring" in the revised manuscript on Page 3, Paragraph 3. We have also added the sentence "Moreover, the 2-OH group was essential for the formation of the bridged hemiacetal or hemiaminal intermediates to ensure the racemization process for the DKR transformation (*vide infra*)." at the end of Paragraph 3 on this page.

- d) Reviewer's comment: "Only Tosyl type amines were evaluated. Did the authors have the results of any other amine?"

Our Response: We have examined a variety of amine / amide substrates including phenylamine **r1**, benzyl amine **r2** and benzamide **r3** in this transformation (Figure R3). Unfortunately, none of these amine / amide substrates provides the target nitrile products at the current stage. This information has been added at the end of Paragraph 4 on Page 3 of the revised manuscript.

unsuccessful amine / amide substrates:

Figure R3. Unsuccessful amine / amide substrates.

- e) Reviewer's comment: "What are the *er* values of axially chiral molecules (e.g., ligand 7, 15, 16, 19) used as catalysts in the asymmetric transformations depicted in Fig. 4? Can the optical purities of the corresponding products be further improved when using catalysts with higher enantiopurities? From 3a to 10, the enantiopurity was visibly increased. Could the authors give any explanation?"

Our Response: We have added the er values of the axially chiral molecules **7**, **15**, **16** and **19** in Fig. 4c in the revised manuscript.

We have chosen the catalytic reaction using the chiral biaryl catalyst **16** as the example to investigate the relationship between the optical purities of the catalyst and the product (Figure R4). The optical purities of the product **28** can be visibly improved when using the catalyst **16** with a higher enantiopurity. This information has been added in Fig. 4c and on Page 8, Paragraph 2 in the revised manuscript.

Figure R4. Applications of catalyst **16** for Asymmetric Reactions.

We have repeated the synthetic transformation from **3a** to **10**, and found that the enantiopurity of the product **3a** was actually remained the same as the starting material **3a**. We have rectified this error in Fig. 4a and on Page 6, Paragraph 2 in the revised manuscript. We are really sorry for this mistake and appreciate to this reviewer for his valuable comments.

- f) Reviewer's comment: "A quite similar work on atroposelective transformation of "Bringmann lactone" derivatives by Wang and co-workers should be cited (J. Am. Chem. Soc. 2016, 138, 6956–6959). Meanwhile, the first work on atropo - enantioselective ring opening of achiral biaryls by Bringmann and co-workers is suggested to be included (Angew. Chem. Int. Ed. 1992, 31, 761–762)."

Our Response: We have added these references as ref. 43 and ref. 41 in the revised manuscript.

- g) Reviewer's comment: "Other minor points to be corrected: Fig. 1, "central-chiral" to "centrally-chiral"; "versatile" to "versatile"; Fig. 2, the chemical bond was pointed to the middle of OH in some structures; please correct; Fig. 3b, for the investigation of the configurational stability of **3a**, solvent "mesitylene" is suggested to be added."

Our Response: We have changed "central-chiral" to "centrally-chiral" and "versatile" to "versatile" in Fig. 1. We have revised the wrong –OH bonds in Fig. 2. We have added the solvent "mesitylene" in Fig. 3b.

Reviewer #2 has recommend publication of this paper after certain revisions on the DFT studies.

- a) Reviewer's comment: "The computed barrier of 33.1 kcal/mol for the loss of the methansulfinate anion (model_TS2) is too high to be reasonable under the reaction conditions. The authors could easily verify this by means of kinetic simulations, and should discuss in more detail the most likely error sources in their calculations (see points 3 and 4)."

Our Response: We thank the reviewer for pointing this out and agree that for the energetic span of > 30 kcal mol⁻¹, reaction at ambient temperature is kinetically difficult. This could be due to the artefact of the model system and we carried out the energetic span investigation of the full system focusing on the key step (see responses to c)). We found that for the full system, the energetic span for the formation of the major product is 23.7 kcal mol⁻¹. This gives a rate constant of 5 s⁻¹ assuming first-order kinetic (elementary step), and a half-life of about 4 hours. This is consistent with good reactivity at 30°C reaction temperature. We also investigated the dependence of energy values on the addition of diffuse functions to basis sets and found no significant changes in the energy values with or without diffuse functions (see response to d)). To clarify the energy span of the full system, we added in the manuscript the following on Page 4, Paragraph 2:

*“For the full system, we found that the reaction pathway leading to the major product via the transition state, **major_TS2**, has an energetic span of 23.7 kcal mol⁻¹, whereas the energetic span for the minor product formation via **minor_TS2** is 27.2 kcal mol⁻¹ (Fig. 3a) and Supplementary Fig. 6). This barrier difference of 3.5 kcal mol⁻¹ translates to an enantiomeric excess of 99%, at experimental temperature of 30 °C. This is in good agreement with experimental observations. In addition, the energetic span of 23.7 kcal mol⁻¹ for the major atropoenantiomer formation is consistent with excellent reactivity at the experimental temperature of 30 °C.”*

- b) **Reviewer’s comment:** *“In Figure 2, the product is higher in energy than the intermediate model_INT2. If this is the case, what is the driving force for the reaction?”*

Our Response: We thank the reviewer for raising this important point and we took careful investigation of the sources of possible errors. We note that in ORCA 4.2.1, which was employed for our calculations, the solvent correction uses the solvent field matching the HF density and does not take into account of the changes due to couple-cluster correlation and this has been reported to alter final energies wildly (see ORCA forum discussion at <https://orcaforum.kofo.mpg.de/viewtopic.php?f=8&t=7476>; login required). In our revision here, we *recalculated* all single-point energies using the recently released **ORCA 5.0.1** version, which allows accurate correction of solvation energies using coupled cluster calculations with SMD model. In this recalculated energy profile for the model reaction (Supplementary Figure 4), the product formation is overall exergonic, allowing the catalytic cycles to proceed. This exergonicity is similarly observed for the full system (Supplementary Figure 6).

- c) **Reviewer’s comment:** *“A possible error source in the calculations is the model system used by the authors in their preliminary study (Figure 2). To address this point, the authors should provide the reaction barrier for the loss of the methanesulfinate anion computed using the full system. I could not find this number anywhere in the SI.”*

Our Response: We thank the reviewer for raising this point. Based on the model system, we further carried out the investigation of the energetic span of the full system based on the rate-determining and stereo-determining mesylate/tosylate elimination step. The results are shown in Supplementary Figure 6, where we found that “The energetic span for the rate-determining TS leading to the major product is 23.7 kcal mol⁻¹ and to the minor product is 27.2 kcal mol⁻¹. This barrier difference of 3.5 kcal mol⁻¹ translates to an enantiomeric excess of 99%, at experimental temperature of 30 °C, which is in good agreement with experimental observations. In addition, the energetic span of 23.7 kcal mol⁻¹ is consistent with excellent reactivity at experimental temperature of 30 °C”. We include the following elaboration in the manuscript on Page 4, Paragraph 2:

*“For the full system, we found that the reaction pathway leading to the major product via the transition state, **major_TS2**, has an energetic span of 23.7 kcal mol⁻¹, whereas the energetic span for the minor product formation via **minor_TS2** is 27.2 kcal mol⁻¹ (Fig. 3a) and Supplementary Fig. 6).”*

- d) **Reviewer’s comment:** *“Another possible error source is basis set. The authors did not include any diffuse basis function in their study (I assume that this was done for efficiency reasons). As the key step of the reaction involves charged species (e.g., the methanesulfinate anion), this choice might lead to errors that are difficult to predict. The authors could verify this by computing (for the smaller model system) the reaction barrier with and without augmented functions.”*

Our Response: We thank the reviewer for this suggestion. Although computationally expensive, we carried out the DLPNO-CCSD(T) at complete basis set (CBS) extrapolation using the aug-cc-pV(DT)Z basis set augmented with diffuse functions for the model system. Due to the linear dependency errors of diffuse functions under “AutoAux” in ORCA software, DLPNO-CCSD(T) was run separately with aug-cc-pVDZ or aug-cc-pVTZ basis set with corresponding auxiliary basis sets aug-cc-pVD(T)Z/C and the obtained values are extrapolated manually according to the following formulae:

$E_{\text{SCF}}^{(X)} = E_{\text{SCF}}^{(\infty)} + A \exp(-\alpha\sqrt{X})$	Eq (1)
$E_{\text{corr}}^{(\infty)} = \frac{X^\beta E_{\text{corr}}^{(X)} - Y^\beta E_{\text{corr}}^{(Y)}}{X^\beta - Y^\beta}$	Eq (2)

for the extrapolation of HF energy (Eq (1)) and of correlation energy (Eq (2)) to the basis set limit, respectively. $E_{SCF/corr}^{(X)}$ is the SCF/correlation energy calculated with basis set of cardinal number X , and $E_{SCF/corr}^{(\infty)}$ is the basis set limit SCF/correlation energy and A , α , and β are constants. For correlation energy, X and Y are the cardinal numbers of the basis sets used for extrapolation ($X=2$, $Y=3$ herein). For Extrapolate(2/3, cc), $\alpha=4.42$, and $\beta=2.46$ and for Extrapolate(2/3, aug-cc), $\alpha=4.3$, and $\beta=2.51$.

We found that the values obtained with aug-cc-p(VT)Z all fall within 1.5 kcal mol⁻¹ of the values obtained with cc-p(VT)Z (Supplementary Figure 4). Furthermore, the relative energies/energy barriers as well as the rate-limiting steps remain unchanged for either basis set. Thus, the addition (or the omission) of diffuse functions does not cause significant value differences; there are no changes in conclusion with regards to mechanism of the reaction.

- e) **Reviewer's comment:** "In model_INT2, the negative charge is most likely delocalized over the oxygen atoms, while the positive charge is delocalized all over the ring. Thus, the Lewis shown in Figure 2 does not represent accurately the electronic structure of the intermediate."

Our Response: We carried out the Natural Bond Orbital (NBO) charges analysis for model_INT2 and found that the NBO charge on N atom is -1.07, and the NBO charges on the O atoms are -1.026 and -1.029. The resonance structures are shown below:

As the NBO charges are similar, it suggests that the negative charge is delocalized over both the N atom and the two O atoms. In our revision, we now show the possible resonance structures and also add in the NBO charges values on the heteroatoms (N and O) for clarity in Fig. 2 of the main text.

- f) **Reviewer's comment:** "The authors should clarify whether they have used the (T_0) approximation in DLPNO calculations, which neglects the couplings between different triples by the off-diagonal Fock matrix elements, or the iterative algorithm (T_1), which is also available for DLPNOCCSD(T)."

Our Response: We thank the reviewer for pointing this out. We have included in the "Computational Methods" section in the Supplementary Information this detail in Paragraph 2:

" T_0 approximation which neglects the couplings between different triples by the off-diagonal Fock matrix elements, instead of the recently published iterative T_1 algorithm, was employed."

- g) **Reviewer's comment:** "The authors should clarify which PNO settings they have used for the DLPNO-CCSD(T) calculations (NormalPNO or TightPNO)"

Our Response: We have included this detail in the "Computational Methods" section in the Supplementary Information this detail in Paragraph 2:

"The NormalPNO settings with $T_{cutPairs} = 10^{-4}$, $T_{cutDO} = 10^{-2}$, $T_{cutPNO} = 3.33 \times 10^{-7}$ and $T_{cutMKN} = 10^{-3}$ was used throughout."

- h) **Reviewer's comment:** "The authors should provide cc-pVDZ and cc-pVTZ values alongside with the extrapolated ones in the SI."

Our Response: We thank the reviewer for this suggestion and have included this in the Supplementary Table 5 and Supplementary Table 6 with appropriate captions:

“Supplementary Table 5. Raw energy values obtained at SMD(toluene)-DLPNO-CCSD(T)/cc-pV(DT)Z basis sets and the complete basis set (CBS) extrapolation. Final single-point (SP) energy = Extrapolated SCF energy + Extrapolated correlation energy. $\alpha = 4.42$ and $\beta = 2.46$ in the extrapolation of SCF and correlation energies. All values have the units of a.u.” and

“Supplementary Table 6. Raw energy values obtained at SMD(toluene)-DLPNO-CCSD(T)/aug-cc-pV(DT)Z basis sets and the complete basis set (CBS) extrapolation. Final single-point (SP) energy = Extrapolated SCF energy + Extrapolated correlation energy. $\alpha = 4.3$ and $\beta = 2.51$ in the extrapolation of SCF and correlation energies. All values have the units of a.u.”

Reviewer #3 has recommend publication of this paper after minor revisions.

- a) **Reviewer's comment:** “Page 1, line 10-11, the sentence “Unfortunately, simultaneous control of the atropoenantioselectivity in such reactions has not been realised with current methods such as transition metal-catalysed asymmetric coupling reactions.” is a bit confusing, please rephrase.”

Our Response: We have rephrased this sentence as “Unfortunately, effective control of the atropoenantioselectivity in the formation of the stereogenic C(sp²)-C(sp²) axis of the axially chiral benzonitrile products has not been realised.”.

- b) **Reviewer's comment:** “The stability of the represent axially chiral benzonitriles and its amine derivative (compound 11) is highly recommended to be investigated by reporting their half time.”

Our Response: We have stirred the mesitylene solution of the axially chiral benzonitrile **3a** and the amine **11** at 180 °C for over 12h. The er values did not change at all. Therefore, it is quite a challenging task to report the half-life time of these axially chiral molecules since they are very stable and the changing rate of the compound er values with different temperatures cannot be obtained at the current stage.

- c) **Reviewer's comment:** “This related work should be cited (Org. Lett. 2018, 20, 6284–6288).”

Our Response: We have added this reference as ref. 47 in the revised manuscript.

- d) **Reviewer's comment:** “SI, page 16 and page page 58, please control the amount of product and the corresponding yield.”

Our Response: We have re-examined the yields and the amounts of products reported on Page 16 and Page 58 in the Supplementary Information. The mistake with the compound **25** has been rectified on Page 16 in the revised Supplementary Information.

- e) **Reviewer's comment:** “For the supplementary: “additional X-ray crystallographic data for compound 11”, inside Alert level C, the Friedel Pair Coverage was indicated to be too low (62%), which maybe not sufficient for the absolute configuration determination, it is recommended to conduct the X-Ray analysis by using copper target or other technologies.”

Our Response: We have re-prepared the single crystals of the compound **11** and subjected them to X-ray analysis using copper target. The updated information (CCDC No. 2116213 and checkcif report) on the crystal structure has been provided in the revised manuscript and Supplementary Information.

REVIEWERS' COMMENTS

Reviewer #1 (Remarks to the Author):

The authors have addressed all my concerns in the revisions and the revised manuscript becomes suitable for publication.

Three non-scientific points need to be correct:

Figure 4, compound 10, insert a space between yield and er value.

Unify the format for kcal mol⁻¹ and kcal/mol in Figure 3.

Page 2, the first sentence after Reaction development, '1a ...and ... 2a were selected as the model substrates'.

Reviewer #2 (Remarks to the Author):

The authors properly addressed all of my comments.

Reviewer #3 (Remarks to the Author):

All of my critiques from last round of review have been well addressed by the authors, and I do not see an problem with publication.

All the 3 reviewers have recommended publication of this manuscript on Nat. Commun., and the Reviewer #1 has pointed out several minor issues:

Reviewer #1

a) Figure 4, compound 10, insert a space between yield and er value.

Our response: We have rectified this error.

b) Unify the format for kcal mol^{-1} and kcal/mol in Figure 3.

Our response: We have rectified this error.

c) Page 2, the first sentence after Reaction development, "1a... and 2a... were selected as the model substrates".

Our response: We have rectified this error.